# A Multi-Year Data Set of Beach-Foredune Topography and Environmental Forcing Conditions at Egmond aan Zee, The Netherlands

**Gerben Ruessink** \*, **Christian S. Schwarz**, **Timothy D. Price** and **Jasper J. A. Donker**

Department of Physical Geography, Faculty of Geosciences, Utrecht University, P.O. Box 80.115,
3508 TC Utrecht, The Netherlands; c.s.schwarz@uu.nl (C.S.S.); t.d.price@uu.nl (T.D.P.);
j.j.a.donker@uu.nl (J.J.A.D.)
\* Correspondence: b.g.ruessink@uu.nl; Tel.: +31-30-2532780

**Abstract:** Coastal dunes offer numerous functions to society, such as sea defense and recreation, and host unique habitats with high biodiversity. Research on coastal dune dynamics has traditionally focused on the erosional impact of short-duration (hours to days), high-wave storm events on the most seaward dune, called the foredune. In contrast, research data on its subsequent slow (months to years), wind-driven recovery are rather rare, yet essential to aid studying wind-driven processes, identifying the most relevant wind-forcing conditions, and testing and improving dune-growth models. The present data set contains 39 digital elevation models and 11 orthophotos of the beach-foredune system near Egmond aan Zee, The Netherlands. The novelty of the data set lies in the combination of long-term observations (6 years; January 2013 to January 2019), with high temporal (intervals of 2–4 months) and spatial resolution ($1 \times 1$ m) covering an extensive spatial domain (1.4 km alongshore). The 25-m high foredune eroded substantially in October 2014, with a maximum recession of 75 $m^3/m$, and subsequently recovered with a rate of approximately 15 $m^3/m/yr$, although with substantial alongshore variability. The data set is supplemented with high-frequency time series of offshore wave, water level, and wind characteristics, as well as various annual subtidal cross-shore profiles, to facilitate its future application in coastal dune research.

**Dataset:** The data set is stored on the Zenodo repository: https://doi.org/10.5281/zenodo.2635416.

**Dataset License:** Creative Commons Attribution 4.0 International.

**Keywords:** foredune; dune erosion; dune growth; aeolian recovery; embryo dunes; beach; storms; remote sensing; Egmond aan Zee

---

## 1. Summary

The beach-foredune system forms a highly dynamic coastal environment that is shaped by the interaction between waves, wind, and vegetation [1,2]. Our extensive knowledge of how storm waves and associated processes [3,4] erode dunes has facilitated the development and application of reliable process-based coastal-erosion models [5] in scientific and applied storm-impact studies [6–9]. In contrast, our understanding of the subsequent slow (months to years) aeolian dune recovery and growth is largely conceptual [10–14]. This is at least in part due to a lack of adequate multi-annual, high resolution (months) topographic data sets of beach-foredune systems to aid studying aeolian processes, to identify wind events most relevant to dune recovery, and to test and improve dune-growth models that are currently being developed [15–18].

Two main types of foredune erosion and recovery can be distinguished based on pre-storm foredune height relative to storm water level. (1) Foredunes in low-wave environments are generally only a few meters high [10,19], which results in overwash and destruction during severe storms [12,20–23]. Based on 10 years of cross-shore profile data collected in the low dune systems of Galveston and Follets Island, Texas, Morton et al. [12] identified four stages in post-storm recovery: beach widening, backbeach aggradation, dune formation, and dune expansion. Using these data and several other data sets collected along the Gulf of Mexico, Houser et al. [13] illustrated how the temporal change in post-storm foredune height, taken as a proxy for foredune recovery, follows a sigmoidal growth curve. The fastest growth takes places several years after the storm, once the beach has widened sufficiently to support significant onshore aeolian transport and vegetation has re-established on the backbeach, facilitating the trapping of the wind-blown sand and hence vertical dune growth. (2) Foredunes in high-wave environments are generally higher than in low-wave environments, with maximum heights between 10 and 30 m [10,19], and their front face is therefore scarped during storms with sufficiently elevated water levels [7,24–26]. During post-storm recovery wind-blown sand is deposited at the base of the nearly vertical front face, which can grow into a dune ramp [14,25,27,28]. In contrast to the situation in low-wave environments, the aeolian recovery can commence immediately because the beach is rather wide after an erosion event due to the deposition of the eroded sand [28]. Embryo dunes may develop on the seaward edge of the ramp following the establishment of vegetation [14,29]. Only once the ramp has developed to a sufficient height can wind-blown sand reach the upper part of the foredune.

The aim of this paper is to present a data set of digital elevation models (DEMs) and orthophotos of the beach-foredune system near Egmond aan Zee, The Netherlands, a high-wave storm-dominated site with an approximately 25-m high foredune. The elevation data set combines a long duration (six years) with a high temporal resolution (typically 2–4 months) and is spatially extensive (1.4 km alongshore) with a high spatial (1 m) resolution. To facilitate the future development and testing of coastal dune evolution models, we supplement the data set with high-frequency time series of offshore wave, water level, and wind characteristics as well as several subtidal bathymetries. The elevation data were collected in the framework of the project "Aeolus meets Poseidon: wind-blown sand transport on wave-dominated beaches" carried out by staff and students from the Department of Physical Geography, Utrecht University, The Netherlands.

## 2. Data Description

### 2.1. Study Site

Egmond aan Zee is located on the approximately 120-km long, uninterrupted, North–South oriented Dutch Holland coast (Figure 1). It faces the semi-enclosed North Sea and is a microtidal, storm-wave dominated site. The annual mean offshore significant wave height $H_{m0}$ and period $T_{m02}$ are about 1.3 m and 4.5 s, respectively. During winter, the monthly mean $H_{m0}$ is substantially higher than in summer (1.8 versus 0.9 m) [30]. During northwesterly storms, $H_{m0}$ can increase to over 7 m. The tide is semi-diurnal, with a neap and spring tidal range of approximately 1.4 and 1.8 m, respectively. Storm surges can raise the water level by more than 1 m, especially when the wind is from the northwesterly to northerly directions. The most frequent winds are, however, from the southwest. The gently ($\approx$1:40) sloping intertidal beach often contains 1 or 2 slipface ridges [31]. Landward of the high-tide level, the profile becomes steeper, and at an elevation of around 3 m above mean sea level (MSL), it changes into the steep (1:2.5) front face of the foredune. At 14 to 17 m + MSL, the profile shows an abrupt change in slope and continues gently to the foredune crest at a height of 20 to 25 m + MSL. Especially this latter, more gently sloping part of the foredune is densely covered in European marram grass (*Ammophila arenaria*). The steep foredune slope has resulted from earlier dune erosion events, with the change in slope marking the location to which the foredune eroded by means of rotational failure [7]. Alongshore variability in foredune shape and height is small. During multiple

years without dune erosion, embryo dunes can develop at the toe of the foredune [7]. The well-sorted quartz sand at the study site has a medium grain size of 250–300 μm, with a tendency to decrease in the landward direction.

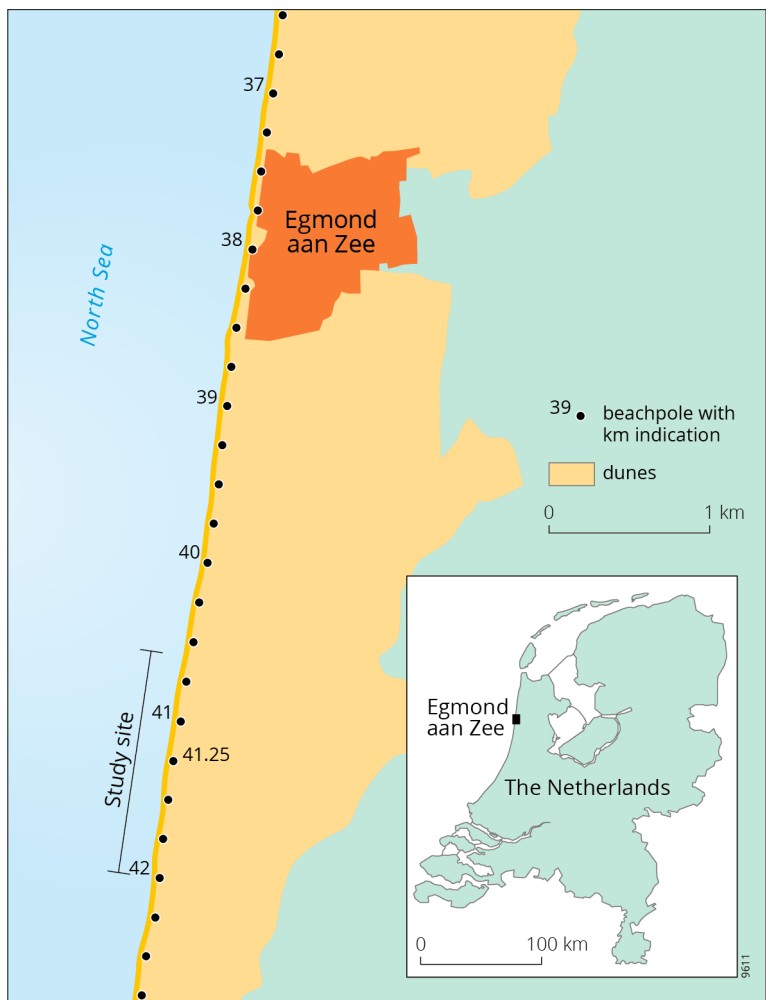

**Figure 1.** Location of study site. The beach poles form an alongshore reference line, with the km number referring to the distance to the zeropoint at the northern end of the Holland coast. The origin of the local coordinate system used here is beach pole 41.25, with positive *x* and *y* in the seaward and southern direction, respectively.

*2.2. Data Records*

The core of the data set in the Zenodo repository is formed by 39 digital elevation models (DEMs) and 11 orthophotos collected from 14 January 2013 to 7 January 2019 (Table 1). The DEMs were computed from 3D point clouds obtained with four different remote-sensing techniques: airborne laser scanning (ALS, 6 surveys), mobile terrestrial laser scanning (MLS, 29 surveys), laser scanning from an unmanned aerial vehicle (UAV-Lidar, 2 surveys), and UAV-acquired structure-from-motion photogrammetry (UAV-SfM, 11 surveys); see Table 1. Eight of the UAV-SfM surveys were performed on the same day as an MLS survey (Table 1). Because the UAV was flown over the northern part of the study area only, the resulting 8 UAV-SfM DEMs are not included here, but the accompanying orthophotos are. The ALS, MLS, and UAV-Lidar point clouds were all measured in the European Terrestrial Reference System 1989 (ETRS89) and then transformed to the Dutch Amersfoort/RD New coordinate system (EPSG:28992) using RDNAPTRANS$^{TM}$2008. Its vertical datum, called NAP, is about equal to MSL. Next, the horizontal RD coordinates were transformed to a local coordinate scheme used in earlier studies at the study site [28,32,33]. In this local scheme, the cross-shore *x* coordinate

is positive onshore and the alongshore $y$ coordinate is positive to the south, with the $xy$ origin being a beach pole (4125 L00) in the study area (RD: 102,572 m, 511,553 m). The angle of rotation between the positive $x$ axis in the RD and local schemes is $177°$. All DEMs cover a (cross-shore × longshore) $300 × 1400$ m area with $x = -250 \cdots 50$ m and $y = -650 \cdots 750$ m, and a 1-m square grid resolution. The UAV-SfM 3D point clouds were initially computed in an arbitrary coordinate system [34,35] and then geo-referenced to the local scheme. The 11 UAV-derived orthophotos provide a visual account of the northern part of study site ($x = -250 \cdots 50$ m and $y = -650 \cdots 100$ m) with a 0.1-m square grid resolution. This higher grid resolution was chosen for improved visualization purposes.

**Table 1.** Overview of available topographic surveys, orthophotos, and subtidal bathymetry.

| # | Date | Type | Photo | Bathymetry |
|---|------|------|-------|-----------|
| 1 | 2013-01-14 | ALS | | T |
| 2 | 2013-04-29 | UAV-SfM | X | |
| 3 | 2013-10-04 | UAV-SfM | X | |
| 4 | 2013-12-10 | MLS | X | |
| 5 | 2014-01-18 | ALS | | |
| 6 | 2014-03-17 | MLS | X | T |
| 7 | 2014-10-10 | MLS | X | |
| 8 | 2015-01-16 | MLS | | |
| 9 | 2015-03-15 | ALS | | |
| 10 | 2015-04-17 | MLS | X | T |
| 11 | 2015-06-29 | MLS | | |
| 12 | 2015-09-29 | MLS | | D |
| 13 | 2015-10-09 | MLS | X | |
| 14 | 2015-10-29 | MLS | | |
| 15 | 2015-12-14 | MLS | | |
| 16 | 2016-01-25 | MLS | | |
| 17 | 2016-02-16 | ALS | | |
| 18 | 2016-02-29 | MLS | | T |
| 19 | 2016-04-18 | MLS | X | |
| 20 | 2016-06-09 | MLS | | |
| 21 | 2016-07-07 | MLS | | |
| 22 | 2016-10-07 | MLS | | |
| 23 | 2016-11-28 | MLS | X | |
| 24 | 2017-01-26 | MLS | | |
| 25 | 2017-01-27 | ALS | | |
| 26 | 2017-03-03 | MLS | | |
| 27 | 2017-05-09 | MLS | X | T |
| 28 | 2017-09-23 | UAV-Lidar | | D |
| 29 | 2017-10-09 | MLS | | |
| 30 | 2017-10-16 | UAV-SfM | X | |
| 31 | 2017-11-03 | UAV-Lidar | | D |
| 32 | 2017-12-20 | MLS | | |
| 33 | 2018-01-23 | MLS | | |
| 34 | 2018-02-13 | ALS | | |
| 35 | 2018-03-21 | MLS | | T |
| 36 | 2018-07-10 | MLS | | |
| 37 | 2018-09-27 | MLS | | |
| 38 | 2018-11-22 | MLS | | D |
| 39 | 2019-01-07 | MLS | | |

[1] An X implies the availability of an orthophoto on the indicated survey date; [2] An entry in this column stands for the presence of subtidal elevation data, with D representing a $1050 × 1400$ m DEM and T 250-m spaced Jarkus transects. The subtidal data were not always surveyed on the same day as the beach-foredune topography. See Section 2.3 for the precise survey dates.

The DEMs are provided as ASC Arc/Info ASCII grids with filenames YYYYMMDD_METHOD.asc, where YYYY, MM, and DD are the year, month, and day of the survey, and METHOD is either ALS, MLS, UAVLidar, or UAVSfM (see Table 1). Each file has 6 header lines, which in our case read

```
ncols          301
nrows          1401
xllcenter      -250.000
yllcenter      -650.000
cellsize       1.000
NODATA_value   -9999.000
```

Here, `ncols` and `nrows` are the number of columns and rows, respectively, `xllcenter` and `yllcenter` are the center $x$ and $y$ of the lower-left cell, `cellsize` is the grid resolution, and `NODATA_value` is the "missing" data value. The six header files are then followed by the `nrows` lines of elevation (in m MSL) at `ncols` $x$ positions, starting at the center of the lower-left cell. Each line (row) corresponds to a cross-shore profile. The orthophotos are provided in GeoTIFF format; their filenames are YYYYMMDD_UAVSfM.tif.

## 2.3. Supplementary Data

To support future advances in dune erosion and recovery modeling, the DEMs and orthophotos are supplemented with

- hourly offshore significant wave height $H_{m0}$ [m] and period $T_{m02}$ [s];
- offshore water level $\eta$ [m MSL] at 10-min intervals;
- wind speed $w_s$ [m/s] and direction $w_\theta$ [°N] at 10 m above ground level, also at 10-min intervals;
- four DEMs of the intertidal and subtidal bathymetry extending to 9 m water depth (Table 1); and
- annual cross-shore bathymetry transects for six 250-m spaced survey lines extending to 14 m water depth (Table 1).

The time series of wave, water level, and wind data are provided in three space-delimited ASCII files called offshoreWaves.txt, offshoreWaterlevels.txt, and coastalWind.txt, respectively. The data columns are preceded by five columns providing time information for each observation: year, month, day, hour, minutes. Each water level value is the arithmetic average value of high-frequency observations over the previous five and next five minutes. Each wind value is based on high-frequency data collected in the previous 10 minutes. All three files contain the observations from 1 January 2013 to 31 January 2019.

The four DEMs with intertidal and subtidal bathymetry were computed from data clouds collected with an RTK-GPS system. For the intertidal measurements, the RTK-GPS was mounted on a quad bike or to a survey wheel for those parts of the beach where the quad could not drive. For the subtidal measurements, the RTK-GPS was, combined with a single beam echosounder, mounted on a personal water craft [36,37]. The surveys were performed on 11 September 2015, 20 September 2017, 3 November 2017, and 20 November 2018. The initial coordinate system was ETRS89, which was transformed to RD (EPSG:28992) and then the local coordinate scheme. The DEMs cover the region $x = -50 \cdots 1000$ m and $y = -650 \cdots 750$ m and have $1 \times 1$ m square grid cells. They are provided as ASC Arc/Info ASCII grids with filenames YYYYMMDD_bathy.asc. The six header lines read

```
ncols          1051
nrows          1401
xllcenter      -50.000
yllcenter      -650.000
cellsize       1.000
NODATA_value   -9999.000
```

The annual bathymetric profiles, part of the Dutch Jarkus database [38], comprise annual vessel-based soundings in transects perpendicular to an alongshore reference line of beach poles (Figure 1). Here, the soundings from 2013 up to and including 2018 are provided for Jarkus survey transects 4050 ($y = -750$ m), 4075 ($y = -500$ m), and so on to 4200 ($y = 750$ m), with 250 m spacing. The names of the survey transects are in decameter from the alongshore zeropoint at the northern end of the Holland coast. The data set in the Zenodo repository contains the 42 cross-shore profiles (7 transects for 6 years) in individual space-delimited ASCII files, each with two columns. The first column is the cross-shore coordinate, which has the same origin and direction as the local $x$ coordinate used in the DEMs, and the second column is the elevation with respect to MSL. The filenames are YYYY_TRANSECT_bathy.txt, in which TRANSECT is the name of the Jarkus survey transect (4050, 4075, ...). The survey dates were 23 January 2013; 25 April 2014; 23 May 2015; 10 March 2016; 1 May 2017; and 23 March 2018.

*2.4. Foredune Change*

Between January 2013 and January 2019, the foredune along the entire study site was eroded twice. The first dune erosion event (Figures 2 and 3a) was on 5&6 December 2013, during a severe (10 Beaufort) northwesterly storm that in The Netherlands became known as the Sinterklaasstorm; elsewhere, the storm was called Xaver (in Germany), Sven (in Sweden), or Bodil (in Denmark). Its high surge and high waves caused substantial flooding and erosion along many North Sea coasts [39]. The offshore water level at the study site reached 2.93 m + MSL, which has an exceedance frequency of 110 times in 1000 years [40]. Dune erosion volumes, computed by differencing DEMs #1 and 4 between the 2.5 m + MSL contour and the location of the change in slope near the crest of the foredune ($\approx$14 m contour in Figure 3a), varied alongshore between about 5 and 35 m$^3$/m. The second major erosion event took place on 21&22 October 2014 during another northwesterly storm. Its highest offshore water level at the study site was 2.75 m, which has an exceedance frequency of 200 times in 1000 years [41]. Erosion was strongly localized, reaching 75 m$^3$/m near $y \approx -200$ m; it is visible in Figure 3 from the landward retreat of especially the 14 m contour, with the change in slope in Figure 3b now closer to the 16 m contour. Elsewhere, dune erosion volumes were substantially lower. After the second storm, wind-blown beach sand was deposited at the base of the steep and scarped foredune. This is obvious in Figure 3 from the substantially more seaward location of the 2, 4, and 6 m contour in Figure 3b. The first isolated embryo dunes formed in the summer of 2015 (Figure 4a) in response to the establishment of the pioneer species sea rocket (*Cakile maritima*) and sand couch (*Elytrigia juncea* subsp. *boreo atlantica*). In 2016 and especially 2017, isolated embryo dunes grew and merged into an alongshore continuous incipient foredune ridge (Figures 3b and 4b), which by the end of the study period, extended from approximately $y = -200$ m to 600 m. Elsewhere along the study area, embryo dunes did not establish or remained isolated. On the whole, the lower part of the foredune increased in volume after the October 2014 storm by approximately 15 m$^3$/m/yr. The alongshore variability in this number (the interquartile range was about 7 m$^3$/m/yr) is probably related to alongshore beach width variation [28].

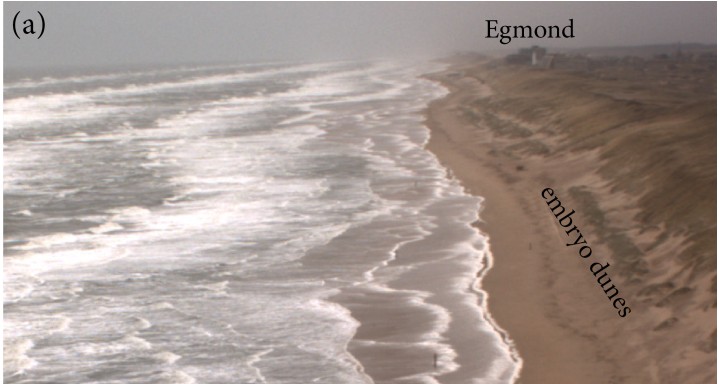

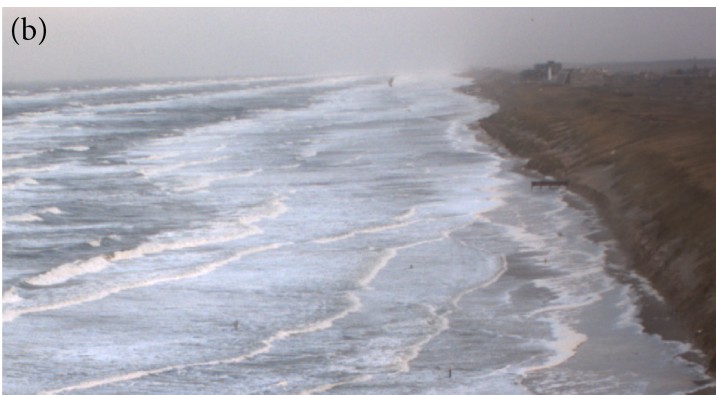

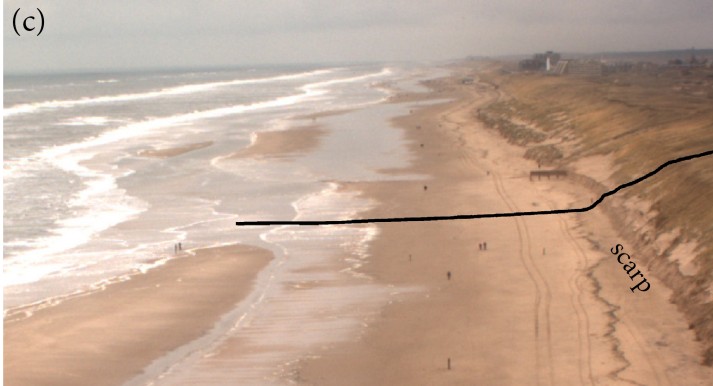

**Figure 2.** Photographs (**a**) prior, (**b**) during, and (**c**) after the Sinterklaasstorm at Egmond aan Zee, taken in the northward (negative $y$) direction by an automated Argus video station [32] on top of an approximately 48 m high tower located on the upper beach at $y = 0$ m. Note that the embryo dunes at the base of the foredune were completely eroded. The drawn line in (**c**) marks the northern end of the study site ($y = -650$ m). The photographs were taken on (**a**) 5 December 09:00 GMT, (**b**) 6 December 08:30 GMT, and (**c**) 7 December 13:30 GMT, 2013. The small vertical black "stripes" on the beach in the lower part of (**c**) are either beach poles or people walking on the beach.

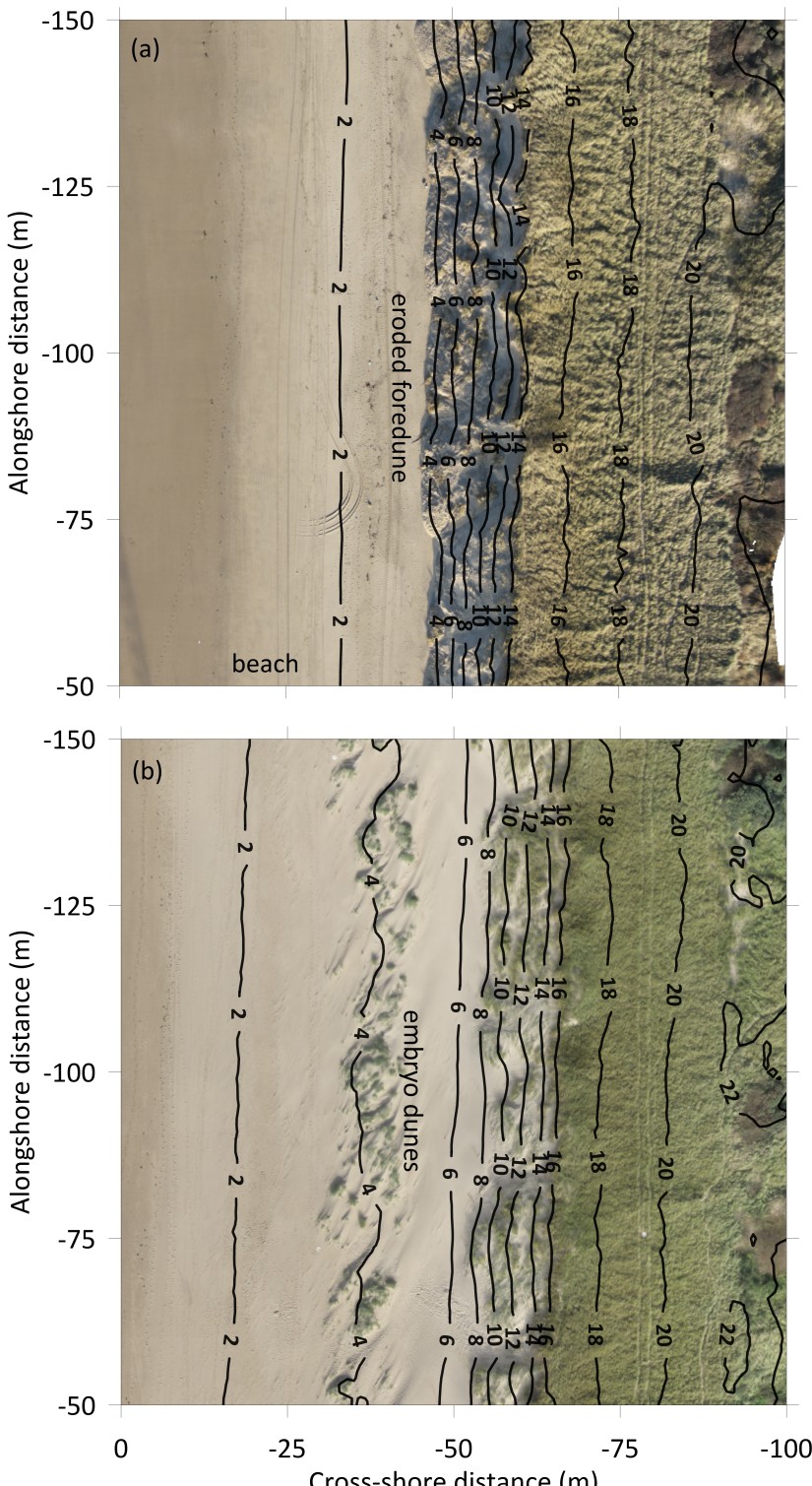

**Figure 3.** Orthophoto of a small part of the study area ($x = -100 \cdots 0$ m, $y = -150 \cdots -50$ m) for (**a**) 12 December 2013 and (**b**) 16 October 2017, with superimposed elevation $z$ contours. North is upward. In (**a**), the contours on the beach and the eroded foredune are based on the 12 December 2013 MLS DEM (#4 in Table 1), while the more landward contours are based on the 18 January 2014 ALS DEM (#5). As mentioned in Section 3.1.2, MLS DEMs do not contain elevations landward of the steep, seaward facing slope of the foredune because of shadowing. Therefore, the ALS contours were added here for completeness. The contours in (**b**) are all based on the 16 October 2017 UAV-SfM DEM (#30).

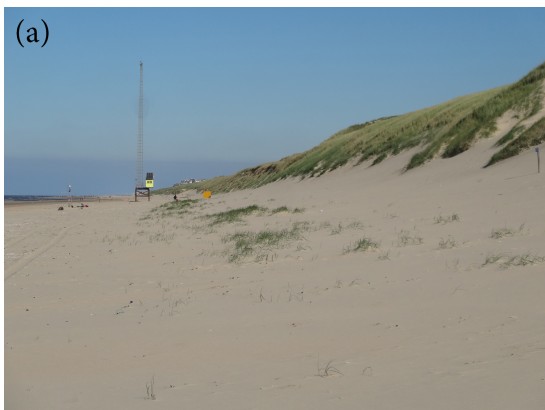
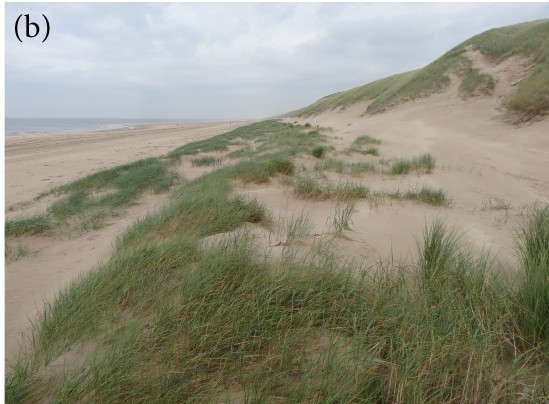

**Figure 4.** Photographs of embryo dunes in the southern part of the study site during autumn (**a**) 2015 and (**b**) 2017. Note the non-vegetated depression between the embryo dunes and the foredune in (**b**); see also Figure 3b. Its effect on MLS data processing is described in Section 3.1.2. The high tower in (**a**) is the Argus video tower from which the photographs shown in Figure 2 were taken. The tower was removed in June 2017 and is hence not visible in (**b**).

## 3. Methods

### *3.1. Topographic Data*

#### 3.1.1. ALS

The ALS point clouds are part of annually collected ALS data sets that span the entire Dutch coast [35,42,43]. The surveys are commissioned by the Dutch governmental institution Rijkswaterstaat, and over the years have been carried out by various commercial contractors with different Lidar systems. Data quality documents, based on data collected over reference areas with known elevation, illustrate that the elevation data has a bias of less than 0.05 m and a standard deviation of less than 0.1 m. The point clouds were computed into $1 \times 1$ m DEMs, with the elevation taken as the average elevation of all points within a 1-m radius around any grid point. Small gaps were filled with the interpolator presented in D'Errico [44]. DEMs of difference sometimes show unrealistically large (2–3 m) annual elevation changes landward of the foredune crest (i.e., landward of the region with significant elevation change by aeolian or marine processes [35,43]). This presumably points to occasional and local failure in the detection of ground values beneath 2 to 3 m high shrubs.

#### 3.1.2. MLS

The 4WD car-mounted MLS system comprised a RIEGL VZ-400 terrestrial laser scanner combined with an OxTS RT3003 inertial navigation system and a dual-receiver GPS navigation system (INS-GPS). The workflow from data collection in the field, subsequent data processing and filtering, and final DEM computation, including the adopted software packages, is described in detail in Donker et al. [28] and not reiterated here. The DEMs generally contain no elevation data landward of the change in profile slope near 14 to 17 m + MSL, as this part of the foredune is shadowed by the steep front face (i.e., not seen by the laser scanner). An error analysis [28] with RTK-GPS points collected on the beach and the lower part of the foredune revealed root mean square differences of about 0.03 m on the beach close to the car, increasing to 0.08 m at 70 m away from the car near the lower non-vegetated part of the foredune, with negligible bias. These observed elevation differences are consistent with those expected from inaccuracies in pitch, roll, and yaw estimated by the INS-GPS. Above 15 m + MSL, the bias becomes positive and the root mean square difference increases, presumably because the laser scanner cannot always see the ground level beneath the dense cover of marram grass. Differencing DEMs #24 (MLS) and #25 (ALS), which were collected only 1 day apart, between the 2.5 m + MSL contour and the

location of the abrupt change in slope revealed an alongshore median volume difference of –0.1 m³/m, with an interquartile range of 1.3 m³/m.

The presence of the embryo dunes from 2016 onward demanded a change to the MLS workflow, as these dunes prevent the laser scanner from viewing the surface of the landward depression in front of the foredune (Figures 3b and 4b). In the original workflow, small data gaps were (just as in the ALS workflow) filled with the D'Errico [44] interpolator. Its application to the extensive alongshore embryo dune-induced shadow zone, however, produced an unrealistic near-horizontal surface from the highest points of the embryo dunes to the foredune. The resulting DEM thus contains too much sand in front of the foredune. To remedy this situation, an alongshore line of points at the deepest part of the depression was added manually to the MLS point cloud prior to gap filling and DEM computation. These additional points allow the interpolator to fill the shadow zone with a depression that is substantially more realistic than the above-mentioned horizontal surface. For MLS surveys in 2016 and 2017, as well as the first two MLS surveys in 2018, the lowest elevation and its cross-shore location for each affected cross-shore transect was taken from the nearest (in time) ALS or UAV-Lidar DEM. From survey #36 (10 July 2018) onward, the elevations and locations of the deepest part of the depression were determined on the same day as the MLS survey with an RTK-GPS mounted on a survey wheel.

### 3.1.3. UAV-Lidar

The two UAV-Lidar surveys were carried out by the Dutch company Shore Monitoring & Research using an AL3-32 lidar system from Phoenix Aerial Systems that, together with an RTK-GNSS system, was attached to a DJI M600pro UAV. The lidar system itself comprised a KVH1725 Fiber Optic Gyro inertial motion unit (IMU), a Velodyne HDL32e laserscanner, a Sony A6000 camera, and a mini Linux computer that stored all data. Processing of the collected 3D point clouds, including ground classification, was carried out with the LASTools software package developed by rapidlasso GmbH; the computation into 1 × 1 m DEMs was identical to the approach described in Section 3.1.1. During both surveys, RTK-GNSS points were additionally measured on the beach for validation of the lidar point clouds. The differences in elevation were generally well below 0.05 m, with negligible bias.

### 3.1.4. UAV-SfM

The 11 UAV-SfM surveys were flown with a fixed-wing Easystar I equipped with a 12.1 Mpixel Canon Powershot D10. About 350 to 1200 collected aerial photographs (Table 2) were processed into a 3D point cloud using the Structure-from-Motion and Multiview-Stereo-View approach [45,46] embedded in AgiSoft Photoscan® Professional Edition. During seven surveys, up to 40 white hexagonal ground control points (GCPs) with black centers were placed on the beach and foredune. The *xyz* coordinates of the centers were measured using RTK-GPS with a horizontal and vertical accuracy of about 0.02 m and 0.04 m, respectively. The centers were later on identified in the images within AgiSoft and used to georeference the point cloud into the local coordinate scheme. During the other four surveys GCPs were not applied. These data were georeferenced using tie points on the foredune without changes in elevation, of which the *xy* coordinates were sampled from individual aerial photos from the March 2014 survey and the *z* coordinate from the March 2014 DEM. These tie points were supplemented with clearly visible beach poles (with known *xy* coordinates), of which the bed elevation was extracted from the MLS point clouds or was measured during the UAV-SfM survey with RTK-GPS. All georeferenced point clouds were processed into 1 × 1 m DEMs as described in Section 3.1.1.

Summary statistics of the *xyz* residuals for the GCPs and tie points, as provided by AgiSoft, include the root mean square error ($\epsilon_{rms}$) for the *x*, *y*, and *z* coordinates separately and the total $\epsilon_{rms}$ (Table 2). For the surveys with GCPs, the $\epsilon_{rms}$ for *x*, *y*, and *z* were typically below 0.05 m, with the error in *z* often being the largest. These errors are about the same as those of the RTK-GPS system used to measure the GCP centers. The total $\epsilon_{rms}$ in these surveys was always less than 0.085 m. For the four

surveys without GCPs, the errors in especially $x$ and $y$ were larger (about 0.1 to 0.15 m), with the total $\epsilon_{rms}$ having values between about 0.11 and 0.21 m. The residuals did not contain a spatial structure, such as a dome-shaped error field reported in some earlier UAV-SfM studies [47].

**Table 2.** Statistics of UAV-SfM surveys.

| Date | #Images | #GCPs | #Tie Points | $\epsilon_{rms}x$ (m) | $\epsilon_{rms}y$ (m) | $\epsilon_{rms}z$ (m) | Total $\epsilon_{rms}$ (m) |
|---|---|---|---|---|---|---|---|
| 2013-04-29 | 636 | | 22 | 0.114 | 0.103 | 0.078 | 0.172 |
| 2013-10-04 | 391 | | 12 | 0.069 | 0.061 | 0.058 | 0.109 |
| 2013-12-10 | 368 | | 24 | 0.112 | 0.155 | 0.081 | 0.208 |
| 2014-03-17 | 428 | 40 | | 0.010 | 0.012 | 0.021 | 0.026 |
| 2014-10-10 | 840 | | 26 | 0.093 | 0.118 | 0.046 | 0.157 |
| 2015-04-17 | 1112 | 40 | | 0.023 | 0.023 | 0.034 | 0.047 |
| 2015-10-09 | 386 | 37 | | 0.041 | 0.022 | 0.028 | 0.055 |
| 2016-04-18 | 833 | 37 | | 0.018 | 0.026 | 0.025 | 0.041 |
| 2016-11-28 | 1045 | 38 | | 0.021 | 0.024 | 0.041 | 0.053 |
| 2017-05-09 | 1179 | 35 | | 0.035 | 0.041 | 0.042 | 0.069 |
| 2017-10-16 | 774 | 22 | | 0.048 | 0.060 | 0.030 | 0.083 |

It is important to mention that the values of $\epsilon_{rms}z$ in Table 2 are based on GCPs and tie points that were located at ground level, with all $z$ thus being the actual bare-earth elevation values. The vertical error in the vegetated parts of the study area, which are located predominantly landward of the change in foredune profile slope (Figure 3), is likely to be larger because photogrammetry-based DEMs contain the top of the vegetation rather than the bare earth beneath the vegetation [34,35,48–50]. It is furthermore likely that this vegetation-induced positive bias changes with the seasons, with the lowest bias in late winter when vegetation cover is rather low and the bare earth is often visible and the highest bias during the growth season when vegetation density is high and the bare earth is thus invisible. We expect a maximum bias of about 0.5 m, the typical height of marram grass on the foredune.

*3.2. Supplementary Data*

3.2.1. Environmental Forcing

Wave data were measured by the "IJmuiden munitiestortplaats (MUN)" wave buoy, located about 40 km west-southwest of Egmond (52°33.000′ N, 004°03.500′ E) in a water depth of 25 m. To obtain a continuous wave time series, small data gaps prior to 1 November 2017 were replaced with measurements from the "Eierlandse Gat (ELD)" wave buoy located 75 km north of the study site (53°16.617′ N, 004°39.700′ E). MUN data gaps from 1 December 2018 to the end of the data record were so severe that the entire period was taken from ELD. Data were absent at MUN and ELD from 1 November 2017 to 20 November 2018. This gap was filled with hourly hindcast waves from the WAVEWATCH-III® model [51], forced with operational NCEP wind fields, for the output grid station closest to Egmond aan Zee (62145; 53°06.150′ N, 002°48.000′ E). The hindcast peak-wave period $T_p$ was converted into $T_{m02}$ as $T_{m02} = T_p/1.33$. The provided water level data were measured at the "IJmuiden buitenhaven" tidal station located 20 km south of the study site (52°27.740′ N, 004°33.289′ E). The few minor data gaps were filled with water level values recorded at the nearby "IJgeul stroommeetpaal" tidal station. Finally, the wind data were collected by the Royal Netherlands Meteorological Institute (KNMI) at the "IJmuiden (WMO 06225)" meteorological station (52°27.733′ N, 004°33.300′ E). The station is located 20 km south of Egmond at the end of the southern IJmuiden harbour mole, close to the local transition from the beach to the foredune. The wind data were measured at 4.4 m above MSL and recomputed (by the KNMI) into values for 10 m above ground.

### 3.2.2. Bathymetry

The quad, survey wheel, and jetski data were collected by Shore Monitoring & Research in cross-shore transects with a 50-m alongshore spacing from the dunefoot to approximately 700 m from the beach; every 250-m spaced Jarkus transect (Section 2.3) was extended to about 1000 m from the beach. The elevation accuracy of the quad, survey wheel, and jetski data are approximately 0.05, 0.03, and 0.10 m, respectively [36,37]. The collected data were processed to grids with a $1 \times 1$ m resolution using point kriging in Surfer® version 16, developed by Golden Software. The Jarkus data have an estimated vertical accuracy of about 0.15 to 0.25 m [38].

## 4. User Notes

The repeat DEMs in our data set can be used to compute DEMs of difference, from which the net volumetric change of the beach-foredune system over a specific time interval can be estimated using a simple integration scheme (e.g., the trapezoidal method). This requires appropriate landward and seaward integration boundaries. We advise using the location of the abrupt change in slope in the foredune's front face as the landward boundary. The MLS DEMs do not contain elevations landward of this boundary because of shadowing (Section 3.1.2), and the UAV-SfM DEMs are here likely to be inaccurate because of a vegetation-induced bias (Section 3.1.4). Earlier analysis of ALS DEMs has indicated that sand deposition is essentially restricted to the lower parts of the foredune [35,43], and we thus expect the error in quantifying volumetric change by neglecting the region landward of the change in slope to be small. The analysis of the total water levels (TWLs) during the time period of interest can provide a suitable definition of the seaward boundary [52]. TWLs can be computed by linearly superimposing the measured offshore water levels and estimates of the wave run-up, for which [53]'s parameterization with the offshore wave data can be applied. The highest TWL values in the time period of interest provide an estimate of the elevation up to which high-wave energy marine processes may have eroded the beach-foredune system and above which morphological change is thus due to aeolian processes. As an example, the 2% exceedance percentile of TWL values between January 2015 and 2019 (the recovery period after the second erosion event) was about 1.7 m, varying between 2.1 m in winter and 1.4 m in summer. To be on the safe side, we chose the 2.5 m contour in Section 2.4 as the seaward integration boundary.

Finally, we note that the monitoring of the beach-foredune system at Egmond aan Zee is scheduled to continue for at least several more years. We intend to update the data in the Zenodo repository through DOI versioning on an annual basis.

**Author Contributions:** Conceptualization and funding acquisition: G.R. and T.D.P.; investigation: G.R. and J.J.A.D.; discussion of results: all authors; writing—original draft preparation: G.R.; writing—review, editing, and visualization: all authors.

**Funding:** This research was funded by The Netherlands Organization for Scientific Research (NWO), grant numbers 13709 and 171.101.

**Acknowledgments:** We thank Rijkswaterstaat and Maarten Zeylmans van Emmichoven (Utrecht University) for providing the ALS data, Marcel van Maarseveen and Henk Markies (Utrecht University) for their support in collecting the MLS data and the UAV aerial photographs, and Marien Boers (Deltares) and Engel Andriessen (KNMI) for providing the Jarkus and wind data, respectively. The wave and water level data were downloaded from the Rijkswaterstaat data portals Waterbase (now obsolete) and Waterinfo (http://waterinfo.rws.nl). The WAVEWATCH-III® hindcast data were downloaded from the WAVEWATCH-III® multigrid production hindcast website at https://polar.ncep.noaa.gov/waves/hindcasts/prod-multi_1.php.

**Conflicts of Interest:** The authors declare no conflict of interest. The funding sponsors had no role in the design of the study; in the collection, analyses, or interpretation of data; in the writing of the manuscript, or in the decision to publish the results.

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
