# Peer review of "A Multi-Year Data Set of Beach-Foredune Topography and Environmental Forcing Conditions at Egmond aan Zee, The Netherlands"

_data, 2013_

Round 1
Reviewer 1 Report
General Comments:
This Data Descriptor paper presents the methodology and description of remote sensing based morphology change data and corresponding environmental forcings for a coastal field site in the Netherlands. The manuscript provides a concise and sufficient overview of the airborne laser scanning (ALS), mobile terrestrial laser scanning (MLS), UAV-lidar, and UAV-structure from motion datasets that were collected along a ~1.4 km stretch of coastline between 2013 and 2019. Across this time period, 39 total high-resolution, 3D topographic datasets were collected. In conjunction with the compiled wave, tide, and wind data and occasional bathymetric information this provides a very valuable dataset for researchers to understand beach and dune recovery processes, as well as storm-induced erosion. The data descriptor is thorough and well written. As such, I have only a few very minor comments below.
Specific Comments:
Line 41: There are many coastal locations in the world with relatively high energy wave climates where the dune elevations do not fall into this specified range of 20-30 m. I would suggest making this statement a bit more generic.
Line 151: Spell out Beaufort Scale.
Line 158: You should define eta.
Section 3.1.4: For all of the other data products it is mentioned briefly that the data were filtered or classified in some form to remove the vegetation. Were there any efforts in the SfM analysis to remove heavily vegetated zones from the output? Or will is there a potential for vegetation artifacts in the bare earth DEM? There is no great way around this, but should just be spelled out how this was handled in the workflow.
Figures: The only real non-text suggestion that I have that would add value a figure that shows an example of each of the various data products. While these data are made public as part of the paper and references are provided to other manuscripts that do include some examples, showing an example visual output product from each of the ALS, MLS, and the two UAV products would help to orient the reader and make it more clear what the scale of features that can resolved from the various collection platforms is. This is a Data Description paper so I think its worthwhile the effort to have a figure that shows a small subset of the actual data.
Author Response
See PDF

Reviewer 2 Report
Ruessink et al., present a probabilistic, data-driven model for studying wind-driven processes, identifying the most relevant wind-forcing conditions, and testing and improving dune-growth. This technique could bring value to erosion forecasting as well as to the development of probabilistic hazard zones. As such, I feel that the research forms an important contribution. The manuscript is well written and describes the motivation, model, and results very clearly. However, I feel the authors need to discuss the limitations of their approach more outright as well as improve some of the structure of the results and discussion sections. Nevertheless, if these comments are adequately addressed, I find that this paper would be a valuable addition to the literature.
Author Response
See PDF
